# DOMAIN-ADAPTIVE IN-CONTEXT GENERATION BENEFITS COMPOSED IMAGE RETRIEVAL

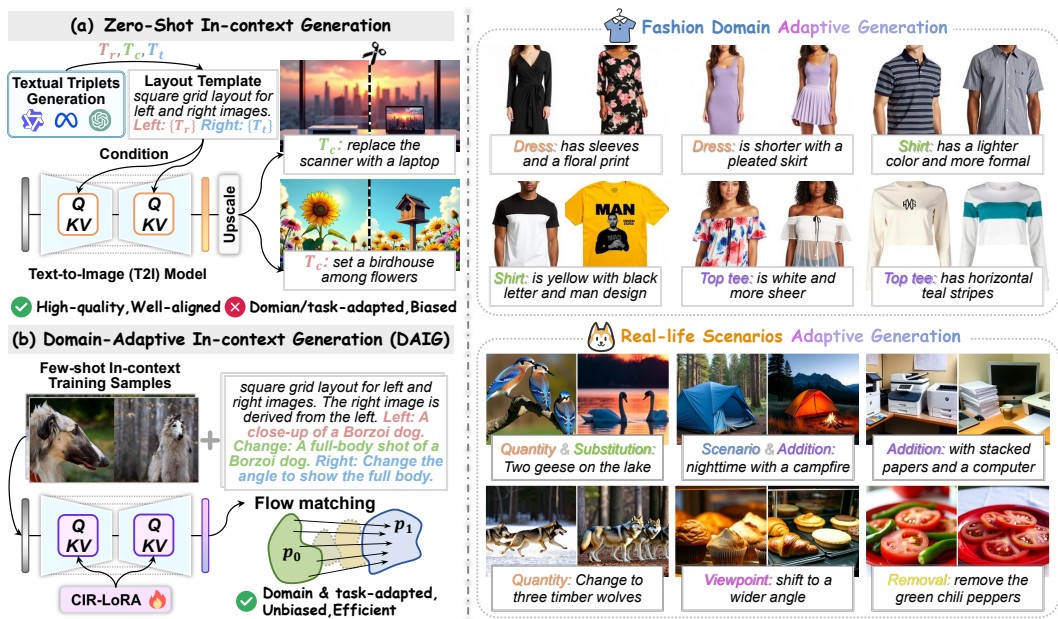

Figure 1: **Motivation of our work.** The left side illustrates: (a) directly leveraging the in-context capability of a pretrained T2I model to generate CIR triplets; (b) our method, which injects domain and task priors into the T2I model from an in-context perspective to improve CIR models in few-shot and fully supervised settings. The right side shows triplets generated by DAIG.

## ABSTRACT

As a vision-language task, Composed Image Retrieval (CIR) aims to integrate information from a bi-modal query (image + text) to retrieve target images. While supervised CIR has achieved notable success in domain-specific scenarios, its reliance on manually annotated triplets restricts its scalability and application. Zero-shot CIR alleviates this by leveraging unlabeled data or automatically collected triplets, yet it often suffers from an intractable domain gap. To this end, we shift the focus to developing robust CIR models under limited labeled data and propose **D**omain-**A**daptive **I**n-context **G**eneration (DAIG), which adapts the in-context capability of a pretrained Text-to-Image (T2I) model to the target domain and the CIR task using few-shot samples and then transforms the LLM-generated textual triplets into unbiased CIR triplets as additional training data. After that, we present a two-stage framework applicable to any supervised CIR approach. The first stage, Distributionally Robust Synthetic Pretraining (DRSP), perturbs visual features to expand the distribution of synthetic data and improve training robustness on it. The second stage, Fine-grained Real-world Adaptation (FRA), fine-tunes on manually annotated triplets by imposing an angular margin on matching pairs to facilitate fine-grained learning. Experiments on two benchmarks validate the effectiveness of our method, *i.e.*, under both few-shot and fully supervised CIR settings, DAIG yields substantial performance gains over CLIP4CIR, BLIP4CIR, and SPRC. The code and data will be released as open source.

# 1 INTRODUCTION

Composed Image Retrieval (CIR) (Vo et al., 2019; Liu et al., 2021; Wu et al., 2021) is a vision-language task that aims to find a target image based on a bi-modal query, which typically consists of a reference image and a relative caption describing the user's intent. Unlike conventional image retrieval (Gordo et al., 2016; Liu et al., 2016), which relies solely on a visual query, CIR enables more fine-grained and intent-driven search by allowing users to express how they want to change or refine the reference image via natural language. In recent years, the emergence of large-scale vision-language pretraining models (VLMs) (Radford et al., 2021; Jia et al., 2021; Li et al., 2023) has drawn increasing attention to CIR, particularly in applications such as e-commerce and multimodal search engines, where image-text compositional queries allow for more flexible retrieval.

Currently, in domain-specific contexts, such as real-life scenarios (Liu et al., 2021) or the fashion domain (Wu et al., 2021), supervised CIR (Baldrati et al., 2022; Liu et al., 2024; Jiang et al., 2024; Xu et al., 2024) has achieved remarkable results by leveraging the powerful multimodal understanding and alignment capabilities of VLMs. However, it heavily relies on meticulously annotated triplets, which makes its application challenging in settings where such labeled triplets are scarce. Therefore, several works have proposed Zero-shot Composed Image Retrieval (ZS-CIR), which can be categorized into, inversion network pretraining (Gal et al., 2022; Saito et al., 2023; Gu et al., 2024b), CIR triplets generation (Gu et al., 2024a; Zhou et al., 2024; Ventura et al., 2024; Levy et al., 2024; Jang et al., 2024) and training-free adaptation of VLMs (Karthik et al., 2024; Yang et al., 2024). However, ZS-CIR is inevitably constrained by the lack of domain-specific contexts, leading to relatively inferior performance compared with supervised counterparts. Motivated by this limitation, we revisit the possibility of few-shot CIR, which leverages limited training samples for supervision to boost CIR model performance with domain-adapted knowledge. Existing few-shot CIR methods (Wu et al., 2023; Hou et al., 2024) typically generate pseudo triplets via prompt tuning or masking techniques. However, such strategies often yield low-quality triplets, providing weak supervision that performs even worse than ZS-CIR counterparts. To address this, prior work Li et al. (2025) designs a zero-shot in-context generation pipeline that generates high-quality triplets using a pre-trained frozen generative model. As illustrated in Figure 1(a), textual triplets are first generated by an LLM and inserted into a layout template, which is then processed by a pre-trained text-to-image (T2I) model to produce semantically related left-right sub-images in a single forward pass. They are subsequently cropped to serve as the reference and target images. However, this approach overlooks the distribution shifts between the generated images and real-world datasets, resulting in the generation of biased triplets that lack domain adaptation.

In this paper, we propose **D**omain-**A**daptive **I**n-context **G**eneration (DAIG), which adapts the in-context capability of pretrained T2I models to the target domain and the CIR task with only few-shot samples. As shown in Figure 1(b), DAIG fine-tunes the T2I model with few-shot samples via a novel parameter-efficient fine-tuning technique, CIR-LoRA. This allows the generative model to preserve its original in-context capabilities while inheriting domain-specific knowledge from the few-shot training samples, thereby producing domain-adapted triplets without bias. With these synthetic triplets, we further design a two-stage CIR training pipeline consisting of, 1) Distributionally Robust Synthetic Pretraining (DRSP), which pretrains CIR models on the synthetic domain-adapted triplets with perturbation applied to the visual features to enhance generalization; 2) Fine-grained Real-world Adaptation (FRA), which fine-tunes on the real-world few-shot training samples by imposing an angular margin for fine-grained pair matching. Extensive experiments on two CIR benchmarks demonstrate that DAIG achieves superior performance over prior state-of-the-art methods. Moreover, DAIG can be seamlessly integrated into existing approaches in a plug-and-play manner, improving retrieval performance without incurring additional inference latency. Furthermore, DAIG is particularly effective in data-scarce scenarios, aligning well with real-world application settings. The contributions of this paper can be summarized as follows:

• We introduce Domain-Adaptive In-Context Generation (DAIG), a novel paradigm that fine-tunes T2I models with only few-shot triplets to provide domain-specific supervision for CIR model. To the best of our knowledge, it is the first work to explore few-shot T2I model tuning for the CIR task.

• We propose a two-stage CIR training pipeline, consisting of Distributionally Robust Synthetic Pretraining (DRSP) and Fine-Grained Real-World Adaptation (FRA), which improve generalization via visual perturbation and adaptation with an angular margin for fine-grained learning.

• Extensive experiments demonstrate that DAIG consistently outperforms recent SOTA methods. Meanwhile, DAIG is plug-and-play on existing methods, effectively boosting performance in few-shot scenarios with no additional inference cost.

## 2 RELATED WORK

**Composed Image Retrieval (CIR)** involves integrating visual and textual information in a composed query to retrieve target images. Most supervised CIR methods (Baldrati et al., 2022; Liu et al., 2024; Xu et al., 2024) leverage the cross-modal alignment capabilities of VLMs, and apply early or late fusion to combine visual and textual information at the query side, achieving impressive performance. Recently, zero-shot CIR has attracted increasing attention, with methods falling into three major categories. The first line of work, such as Pic2Word (Saito et al., 2023) and SEARLE (Baldrati et al., 2023), is based on textual inversion (Gal et al., 2022), where a projection network is trained to map the reference image into a pseudo-word token. LinCIR (Gu et al., 2024b) further improved the efficiency and effectiveness of this paradigm by training the inversion network entirely in the text modality. Second, training-free methods (Karthik et al., 2024; Yang et al., 2024) utilize the powerful reasoning capabilities of LLMs to perform generative retrieval. However, these approaches are constrained by their model complexity and slow inference speed. Finally, several methods attempt to automatically construct triplet data, including synthetic approaches such as CompoDiff (Gu et al., 2024a), VISTA (Zhou et al., 2024), and CoAlign (Li et al., 2025), as well as CoVR-BLIP (Ventura et al., 2024), CASE (Levy et al., 2024), and VDG (Jang et al., 2024), which expand existing image-text datasets to form CIR triplets. These approaches are flexible and can be readily applied within off-the-shell supervised CIR frameworks. However, the constructed triplets are not tailored to a specific domain. Our triplet synthesis pipeline is fine-tuned on a small amount of domain-specific triplet data to improve the performance under supervised and few-shot CIR settings.

**Text-to-Image (T2I) generation** has undergone significant transformations, from early methods based on Generative Adversarial Networks (GANs) (Goodfellow et al., 2020; Arjovsky et al., 2017) to the prevailing diffusion-based models (Sohl-Dickstein et al., 2015; Ho et al., 2020; Dhariwal & Nichol, 2021). With the introduction of latent diffusion models (LDMs) (Rombach et al., 2022), the fidelity between text and image has greatly improved, while also reducing computational costs through latent space operations. This has facilitated high-resolution image generation, such as with Stable Diffusion (Rombach et al., 2022; Podell et al., 2023) and DALL-E 2 (Ramesh et al., 2022). Recently, the integration of transformer architectures (Vaswani et al., 2017) in Diffusion Transformers (DiT) (Peebles & Xie, 2023) has further improved scalability and in-context capabilities, giving rise to several advanced models, including PixArt (Chen et al., 2023), SD3 (Esser et al., 2024), and Flux (Labs, 2024). These models adopt flow matching (Lipman et al., 2022; Liu et al., 2022) as the optimization objective, achieving state-of-the-art generation quality, and supporting multi-object generation as well as multi-subgraph layout. In the CIR domain, CompoDiff (Gu et al., 2024a) and VISTA (Zhou et al., 2024) are the first to leverage T2I models for triplet construction, but their outdated, editing-based approaches produce low-quality images with artifacts and pixel-level bias. More recently, as shown in Figure 1(a), CoAlign (Li et al., 2025) exploits the in-context capability of DiTs to generate semantically related sub-images in a single forward pass, which is more aligned with the CIR task. Our DAIG tackles the limitations of this paradigm in domain-specific scenarios.

## 3 METHODOLOGY

### 3.1 PRELIMINARY

A CIR triplet is denoted as $(I_r, T_c, I_t)$, where $I_r$ is the reference image and $T_c$ represents the relative caption, the two together constitute a composed query corresponding to the target image $I_t$. CIR requires integrating the two modalities within the composed query, and then retrieving $I_t$ from a large image gallery. At present, mainstream supervised approaches leverage the alignment capabilities of VLMs. In addition, they incorporate a modality integration module, and place the query and target features into a joint embedding space optimized through contrastive learning. During inference, for a composed query, cosine similarity is computed with all images in the gallery, and the top-K candidates with the highest similarity scores are returned as the retrieval results.

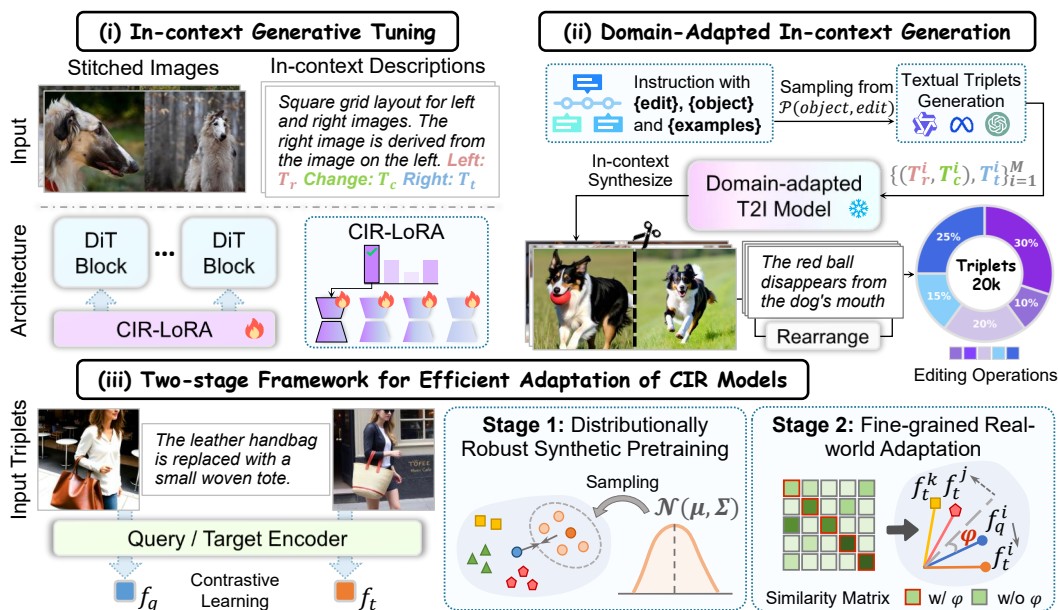

Figure 2: **Overview of our method.** (i) In-context fine-tuning of a T2I model with CIR-LoRA to align with the target domain and inject the CIR task prior. (ii) The DAIG pipeline, which combines the tuned T2I model from i) with an LLM to generate diverse CIR triplets closely aligned with the target domain. (iii) A two-stage CIR framework, which consists of robust pretraining on synthetic triplets and fine-grained adaptation on real-world manually annotated triplets.

## 3.2 DOMIAN-ADAPTIVE IN-CONTEXT GENERATION

**In-context Generative Tuning.** As outlined earlier, zero-shot in-context triplet generation suffers from notable limitations: (1) there exists **an intractable gap** between the generated images and the target domain; (2) **the absence of task priors** causes the left and right sub-images to share overly similar backgrounds, which introduces additional bias when they serve as the reference and target images. As illustrated in Figure 2(i), given a set of target-domain triplets, denoted as $\mathcal{S} = \{(I_r^i, T_c^i), I_t^i\}_{i=1}^N$, we sample a small subset[1] and convert them into an in-context form: (1) concatenate $I_r$ and $I_t$ side by side into a stitched image, which embodies rich visual contextual information of the target domain; (2) employ a captioner to annotate $I_r$ and $I_t$, yielding $T_r$ and $T_t$, which combined with $T_c$, are then inserted into the layout template shown below, resulting in an in-context description, i.e., $T_{ic}$,

> ***In-context Description***: Square grid layout for left and right images. The right image is derived from the image on the left. Left: $T_r$. Change: $T_c$. Right: $T_t$.

Subsequently, we use the obtained in-context pairs to fine-tune a T2I model $\mathcal{G}(\cdot)$. To be specific, the backbone of $\mathcal{G}$ is kept frozen, while a learnable CIR-LoRA module is merged into the cross-attention layer of each block, as depicted below,

$$\text{Attention}(Q, K, V) = \text{Softmax}\left(\frac{QK^T}{\sqrt{d}}\right)V, \quad K = \mathbf{W_k}\tau_{txt}(T_{ic}), \quad V = \mathbf{W_v}\tau_{txt}(T_{ic}), \quad (1)$$

where $Q$ is the image query, and the text encoder $\tau_{txt}$ encodes the in-context descriptions, serving as $K$ and $V$ to inject the textual condition. For each projection weight $\mathbf{W} \in \mathbb{R}^{d \times l}$, CIR-LoRA adopts a Mixture-of-Experts (MoE) (Jacobs et al., 1991) structure, and introduce $K$ learnable experts, i.e, $\mathbf{B}^i \in \mathbb{R}^{d \times r}, \mathbf{A}^i \in \mathbb{R}^{r \times l}$ $i = 1, 2...K$, where $r$ is the rank of each expert, $d$ and $l$ denote the input and output dimensions, respectively. A routing function $\mathcal{R}$[2] then analyzes the characteristics of each in-context description and assigns sample-wise expert weights, i.e., $\mathbf{r} = \mathcal{R}[\tau_{txt}(T_{ic})] \in \mathbb{R}^K$. The

---

[1]Unless otherwise specified, we use 32 instances, which has been shown to be sufficient by our experiments.
[2]The routing function can be simply implemented by an MLP.

updated weight is finally formulated by $\mathbf{W}' = \mathbf{W} + \Delta\mathbf{W} = \mathbf{W} + \beta \cdot \sum_{i=1}^{K} \mathbf{r}^i \cdot \mathbf{B}^i \mathbf{A}^i$, where $\beta$ is a learnable layer-specific scaling factor, and only $\Delta\mathbf{W}$ is trainable. The training process is supervised by flow matching (Liu et al., 2022), and since $r \ll \min(d, l)$, it is highly parameter-efficient, with low computational and time overhead.

Our design offers three main advantages: (1) ***Distribution Alignment***. As an intrinsic property of LoRA (Hu et al., 2021), it can effectively capture and align with the target domain distribution from only a few input samples. (2) ***Task Prior Injection***. The in-context descriptions allow the T2I model to grasp the objective of the CIR task, while the MoE structure assigns optimal expert weights to each input, enhancing adaptability to the wide spectrum of editing operations present in the relative captions. After optimization, a domain-adaptive T2I model $\mathcal{G}'$ is produced, which is able to synthesize unbiased reference and target images that are closer to the target domain distribution, thus enabling the construction of domain-adaptive triplets.

**Domian-Adaptive In-context Generation (DAIG).** Equipped with $\mathcal{G}'$, as shown in Figure 2(ii), we are able to generate additional high-quality training data by constructing in-context descriptions and transforming them into CIR triplets through $\mathcal{G}'$. To this end, we design an instruction template $\mathcal{P}(object, edit)$ to guide an LLM in generating diverse textual triplets, as shown below,

> ***Instruction Template***: Based on the following elements $\{object\}$ and $\{edit\}$, generate a triplet $(T_r, T_c, T_t)$ that conforms to the requirements of CIR. Follow the JSON format for output, with the language style aligned to that of $\{examples\}$.

where $object$ and $edit$ are randomly sampled from predefined sets. These sets can be constructed either manually or automatically based on prior knowledge, and should be diverse and well aligned with the target domain. Further details are provided in the Appendix. $examples$ are sampled from the given target-domain triplets in $\mathcal{S}$ and serve to guide and constrain the language style.

Through iterative sampling of instructions from $\mathcal{P}$ and using them to prompt the LLM, we produce $M$ textual triplets, which are then converted into in-context descriptions and fed into $\mathcal{G}'$ to synthesize $I_r$ and $I_t$, yielding CIR triplets $\mathcal{S}' = \{I_r^i, T_c^i, I_t^i\}_{i=1}^M$. Since, after tuning, $\mathcal{G}'$ rarely outputs low-quality $I_r$ and $I_t$ (e.g., with artifacts or layout errors), unlike CoAlign (Li et al., 2025), we do not require additional data filtering. In summary, our pipeline guarantees that the synthetic triplets are diverse, of high fidelity, and closely aligned with both the target domain and the CIR task. Representative triplets generated by DAIG are shown on the right side of Figure 1. More details, including the complete instruction template and additional visualizations, are provided in the Appendix.

### 3.3 TWO-STAGE FRAMEWORK FOR EFFICIENT ADAPTATION OF CIR MODELS

Our two-stage framework comprises pretraining on synthetic domain-adaptive triplets, followed by fine-grained adaption on the given real-world data. This framework is highly flexible and can be seamlessly integrated with any existing supervised CIR approach, delivering substantial performance gains under both few-shot and fully supervised settings.

**Distributionally Robust Synthetic Pretraining (DRSP).** Images synthesized by a T2I model tend to fall into a sparse distribution with respect to the target domain. Therefore, directly training CIR models on the synthetic domain-adaptive triplets, i.e., $\mathcal{S}'$, often leads to overfitting and suboptimal performance. Motivated by domain generalization (DG) (Zhang et al., 2017; Li et al., 2022) and considering uncertainty[3], we perturb the visual features to broaden the distribution, thus increasing the fitting difficulty and training robustness. Specifically, for the visual features output by the image encoder of a CIR model, denoted as $\mathbf{v} \in \mathbb{R}^{B \times L \times D}$, we hypothesize that their statistics, namely mean and standard deviation, follow a multivariate Gaussian distribution,

$$\mu(\mathbf{v}) \in \mathbb{R}^{B \times D} := \frac{1}{L} \sum_{l=1}^{L} \mathbf{v}_{b,l,d}, \quad \sigma(\mathbf{v}) \in \mathbb{R}^{B \times D} := \sqrt{\frac{1}{L} \sum_{l=1}^{L} (\mathbf{v}_{b,l,d} - \mu(\mathbf{v}))}, \quad (2)$$

where $\mu(\mathbf{v})$ and $\sigma(\mathbf{v})$ determine the domain-specific characteristics of the input images, also referred to as the style (Ulyanov et al., 2016). By computing the variance along the batch dimension as shown

---

[3]During training, with probability $p$, we decide whether to apply perturbation for a batch.

below, we complete the multivariate Gaussian modeling of the statistics,

$$\Sigma_\mu^2(\mathbf{v}) = \frac{1}{B}\sum_{b=1}^{B}(\mu(\mathbf{v}) - \mathbb{E}_b[\mu(\mathbf{v})])^2, \quad \Sigma_\sigma^2(\mathbf{v}) = \frac{1}{B}\sum_{b=1}^{B}(\sigma(\mathbf{v}) - \mathbb{E}_b[\sigma(\mathbf{v})])^2. \tag{3}$$

These multivariate Gaussian distributions allow sampling via the re-parameterization trick (Kingma & Welling, 2013), which produces $\tilde{\mu}(\mathbf{v}) = \mu(\mathbf{v}) + \epsilon_\mu\Sigma_\mu(\mathbf{v})$, $\epsilon_\mu \sim \mathcal{N}(\mathbf{0},\mathbf{I})$ and $\tilde{\sigma}(\mathbf{v}) = \sigma(\mathbf{v}) + \epsilon_\sigma\Sigma_\sigma(\mathbf{v})$, $\epsilon_\sigma \sim \mathcal{N}(\mathbf{0},\mathbf{I})$. Finally, perturbations are applied to $\mathbf{v}$ as follows,

$$\tilde{\mathbf{v}} = \tilde{\mu}(\mathbf{v})\frac{\mathbf{v} - \mu(\mathbf{v})}{\sigma(\mathbf{v})} + \tilde{\sigma}(\mathbf{v}). \tag{4}$$

Subsequently, $\mathbf{v}$ is replaced with $\tilde{\mathbf{v}}$ for subsequent modality fusion and alignment. DRSP perturbs visual features using batch-level samples, broadening the sparse distribution of the synthetic triplet data into a more robust approximation of the target domain. The model is optimized with a standard contrastive loss (He et al., 2020), introduces no additional parameters during training, and disables perturbations at inference, thereby incurring no extra inference cost.

**Fine-grained Real-world Adaptation (FRA).** After pretraining on synthetic triplets, FRA performs fine-grained adaptation on the given manually annotated triplets $\mathcal{S}$, further reducing the domain gap while strengthening the model's capacity for fine-grained discrimination. Specifically, after feature extraction, the cosine similarity between each query feature and all target features within a batch is computed, and after applying softmax, the predicted probability is obtained as follows,

$$p_{i,j} = \frac{e^{cos(\theta_{i,j}/\tau)}}{\sum_{k\in\mathcal{B}} e^{cos(\theta_{i,k}/\tau)}}, \quad \theta_{i,j} = \arcos(\text{sim}(f_q^i, f_t^j)) + \varphi \cdot \mathbb{I}(i=j), \tag{5}$$

where $\text{sim}(\cdot,\cdot)$ is the cosine similarity and $\tau$ is a learnable coefficient used to scale the predicted probability distribution. $\varphi$ multiplied by the indicator function $\mathbb{I}(\cdot)$, imposes an angular margin on all matching pairs (the diagonal entries of the similarity matrix). The predicted $p_{i,j}$ are then used to compute cross-entropy with the ground truth for model optimization. In summary, FRA imposes a margin in the angular space, guiding the CIR model to acquire finer-grained knowledge from manually annotated target-domain data. Along with DRSP, it constitutes two complementary stages that collectively boost CIR performance under both few-shot and fully supervised scenarios.

## 4 EXPERIMENTS

### 4.1 EXPERIMENTAL SETTINGS

**Evaluation Benchmarks.** (1) FashionIQ (Wu et al., 2021), designed to emulate real-world on-line shopping scenarios, focuses on fashion-related images. It comprises 30,134 triplets constructed from 77,684 images. (2) CIRR (Liu et al., 2021), the first open-domain CIR dataset, contains 36,554 annotated triplets, split into training, validation, and test sets with a ratio of 8:1:1. Unlike FashionIQ, CIRR features more diverse real-world scenarios and mitigates issues such as narrow domain coverage and high false negative rates, enabling a more robust and generalizable evaluation.

**Evaluation Metrics.** We adopt $Recall@K$ as the primary evaluation metric, which measures the likelihood of retrieving at least one correct target image within the top-K ranked results. For CIRR, we additionally report $Recall_s@K$, which measures performance within a subset of visually similar candidates associated with each query, offering finer-grained insight under challenging distractor conditions. The CIRR overall performance is measured by $Avg. = \frac{Recall@5 + Recall_s@1}{2}$.

**Implementation Details.** (1) For DAIG, the stitched image resolution for both training and inference is $224 \times 448$. we adopt the T2I model, namely *Flux.1-dev* (Labs, 2024), and fine-tune it with CIR-LoRA (rank=32, two experts) on a single H800 GPU. Regardless of the data rates, we sample only 32 triplets for training, with a total of $15,000$ steps, a learning rate of $1 \times 10^{-5}$ and a batch size of $1$. To construct the in-context descriptions, we employ *Qwen2.5-VL-32B* Bai et al. (2025) as the captioner, while *Qwen2.5-32B* Yang et al. (2025) serves as the LLM for producing $20k$ textual triplets. (3) For DRSP and FRA, the probability $p$ of applying perturbation is set to 0.5 and the angular margin $\varphi = 1 \times 10^{-3}$. All other hyperparameters follow the original settings of the specific CIR approach.

Table 1: **Performance comparison on the FashionIQ validation set.** The best results are marked in bold. † indicates results reproduced by us.

| Rate | Method | Dress | | Shirt | | Toptee | | Average | |
|---|---|---|---|---|---|---|---|---|---|
| | | K=10 | K=50 | K=10 | K=50 | K=10 | K=50 | K=10 | K=50 |
| zero-shot | CoVR-BLIP (Ventura et al., 2024) | 21.95 | 39.05 | 30.37 | 46.12 | 30.78 | 48.73 | 27.70 | 44.63 |
| | CompoDiff (Gu et al., 2024a) | 37.78 | 49.10 | 41.31 | 55.17 | 44.26 | 56.41 | 39.02 | 51.71 |
| 32-shot | CLIP4CIR† (Baldrati et al., 2022) | 20.77 | 42.98 | 26.15 | 43.96 | 28.76 | 48.95 | 25.23 | 45.30 |
| | **+ DAIG (Ours)** | 24.69 | 48.34 | 30.91 | 50.83 | 33.91 | 56.60 | 29.84 | 51.93 |
| | BLIP4CIR† (Liu et al., 2024) | 7.39 | 20.87 | 10.79 | 22.72 | 10.76 | 24.17 | 9.65 | 22.59 |
| | **+ DAIG (Ours)** | 29.95 | 54.64 | 31.31 | 52.45 | 36.46 | 60.43 | 32.57 | 55.84 |
| | SPRC† (Xu et al., 2024) | 26.57 | 47.05 | 28.46 | 48.33 | 31.31 | 55.33 | 28.78 | 50.24 |
| | **+ DAIG (Ours)** | **38.92** | **61.08** | **47.69** | **68.06** | **48.44** | **70.02** | **45.02** | **66.38** |
| 1% | CLIP4CIR† (Baldrati et al., 2022) | 24.79 | 46.75 | 28.85 | 48.04 | 30.80 | 53.65 | 28.15 | 49.48 |
| | **+ DAIG (Ours)** | 24.74 | 49.23 | 31.94 | 51.52 | 34.78 | 57.22 | 30.49 | 52.66 |
| | BLIP4CIR† (Liu et al., 2024) | 14.43 | 33.02 | 17.08 | 32.68 | 18.26 | 36.82 | 16.59 | 34.17 |
| | **+ DAIG (Ours)** | 30.49 | 55.78 | 32.92 | 54.51 | 37.94 | 62.32 | 33.79 | 57.54 |
| | SPRC† (Xu et al., 2024) | 31.88 | 54.93 | 37.05 | 58.05 | 39.98 | 65.73 | 36.30 | 59.57 |
| | **+ DAIG (Ours)** | **39.66** | **61.97** | **47.25** | **68.65** | **49.62** | **71.29** | **45.51** | **67.30** |
| 100% | CLIP4CIR (Baldrati et al., 2022) | 33.81 | 59.40 | 39.99 | 60.45 | 41.41 | 65.37 | 38.32 | 61.74 |
| | BLIP4CIR (Liu et al., 2024) | 42.09 | 67.33 | 41.76 | 64.28 | 46.61 | 70.32 | 43.49 | 67.31 |
| | Re-ranking (Liu et al.) | 48.14 | 71.43 | 50.15 | 71.25 | 55.23 | 76.80 | 51.17 | 73.13 |
| | CaLa (Jiang et al., 2024) | 42.38 | 66.08 | 46.76 | 68.16 | 50.93 | 73.42 | 46.69 | 69.22 |
| | CCIN (Tian et al., 2025) | **49.38** | 72.58 | 55.93 | 74.14 | 57.93 | 77.56 | 54.41 | 74.76 |
| | QuRe (Kwak et al., 2025) | 46.80 | 69.81 | 53.53 | 72.87 | 57.47 | 77.77 | 52.60 | 73.48 |
| | SPRC† (Xu et al., 2024) | 48.39 | 72.09 | 55.15 | 74.39 | 58.08 | 78.38 | 53.87 | 74.95 |
| | **+ DAIG (Ours)** | 49.08 | **73.72** | **56.58** | **75.07** | **59.56** | **79.09** | **55.07** | **75.96** |

## 4.2 QUANTITATIVE RESULTS

**Comparison with state-of-the-art methods.** Compared with recent state-of-the-art approaches, our method achieves superior performance on both the CIRR and FashionIQ benchmarks. As presented in Table 2 and 1, our method achieves SOTA results under both few-shot and fully supervised CIR settings. Specifically, when fine-tuned on full training set, DAIG reaches up to **84.10%** on Recall@5 on the CIRR test set, surpassing the baseline SPRC by **1.88%**. Moreover, DAIG outperforms the Re-ranking by **2.35%** on average R@10, further demonstrating its strong advantage over re-ranking based methods. Similarly, on the FashionIQ validation set, our method outperforms all existing SOTA approaches, surpassing the recent CCIN by **1.20%** in terms of average R@50.

**Plug-and-Play manner.** An appealing property of our method is the plug-and-play nature, which guarantees an effective combination with existing approaches. Without introducing additional inference complexity, our method yields substantial improvements over several widely adopted baselines, including CLIP4CIR, BLIP4CIR, and SPRC. Concretely, under the 32-shot setting on the CIRR test set, our generative pipeline enhances the three methods by **11.59%**, **29.15%**, and **14.80%** in terms of Recall@5, thereby validating the effectiveness and versatility of our approach.

**Generalization across different data rates.** A key strength of our approach is the generalizability across varying proportions of human-annotated triplets used. Specifically, under different settings such as 32-shot, 1% and 100%, our pipeline consistently improves over baseline methods including CLIP4CIR, BLIP4CIR and SPRC. For instance, our method enhances SPRC across all three data rates by **14.80%**, **6.75%**, and **1.88%** in terms of Recall@5 on the CIRR test set, **16.24%**, **9.21%**, **1.2%** on the FashionIQ validation set, demonstrating strong robustness regardless of the data scale.

**Comparison with other few-shot CIR methods.** Since our method is capable of fine-tuning the T2I model with only a minimal number of samples, we compare our method against existing Few-shot CIR approaches under 8-shot and 16-shot settings. Specifically, we evaluate our method against the two available Few-shot CIR methods, PromptCLIP (Wu et al., 2023) and PTG (Hou et al., 2024). As shown in Table 4, DAIG consistently outperforms PromptCLIP and PTG by a large margin under both 8-shot and 16-shot settings, highlighting the effectiveness of DAIG in few-shot learning.

Table 2: **Performance comparison with existing methods on the CIRR test set.** The best results are marked in bold. † indicates results reproduced by us.

| Rate | Method | Recall@K | | | | Recall$_s$@K | | | Avg. |
|---|---|---|---|---|---|---|---|---|---|
| | | K=1 | K=5 | K=10 | K=50 | K=1 | K=2 | K=3 | |
| zero-shot | CoVR-BLIP (Ventura et al., 2024) | 38.48 | 66.70 | 77.25 | 91.47 | 69.28 | 83.76 | 91.11 | 67.99 |
| | CompoDiff (Gu et al., 2024a) | 26.71 | 55.14 | 74.52 | 92.01 | 64.54 | 82.39 | 91.81 | 59.84 |
| | CASE (Levy et al., 2024) | 35.40 | 65.78 | 78.53 | 94.63 | 64.29 | 82.66 | 91.61 | 65.04 |
| 32-shot | CLIP4CIR† (Baldrati et al., 2022) | 22.87 | 52.12 | 64.63 | 88.55 | 52.12 | 73.61 | 87.35 | 52.12 |
| | **+ DAIG (Ours)** | 31.02 | 63.71 | 75.81 | 93.23 | 59.30 | 81.01 | 91.47 | 61.51 |
| | BLIP4CIR† (Liu et al., 2024) | 9.06 | 27.86 | 38.65 | 64.48 | 26.68 | 49.30 | 69.47 | 27.27 |
| | **+ DAIG (Ours)** | 26.75 | 57.01 | 70.19 | 89.83 | 57.16 | 79.23 | 90.99 | 57.09 |
| | SPRC† (Xu et al., 2024) | 29.88 | 57.61 | 69.25 | 88.46 | 67.30 | 84.29 | 92.24 | 62.46 |
| | **+ DAIG (Ours)** | **42.05** | **72.41** | **82.00** | **95.25** | **71.33** | **86.84** | **94.46** | **71.87** |
| 1% | CLIP4CIR† (Baldrati et al., 2022) | 27.61 | 57.90 | 70.12 | 91.18 | 56.77 | 77.16 | 88.70 | 57.34 |
| | **+ DAIG (Ours)** | 32.51 | 64.75 | 77.16 | 94.02 | 60.92 | 82.15 | 91.88 | 62.84 |
| | BLIP4CIR† (Liu et al., 2024) | 12.99 | 35.78 | 48.39 | 75.11 | 37.40 | 62.60 | 80.15 | 36.59 |
| | **+ DAIG (Ours)** | 28.65 | 58.99 | 72.34 | 91.23 | 58.70 | 80.41 | 91.13 | 58.59 |
| | SPRC† (Xu et al., 2024) | 38.65 | 68.77 | 79.42 | 94.36 | 71.93 | 87.04 | 94.10 | 70.35 |
| | **+ DAIG (Ours)** | **44.55** | **75.52** | **84.87** | **96.22** | **74.77** | **89.08** | **95.23** | **75.15** |
| 100% | CLIP4CIR (Baldrati et al., 2022) | 38.53 | 69.98 | 81.86 | 95.93 | 68.19 | 85.64 | 94.17 | 69.09 |
| | BLIP4CIR (Liu et al., 2024) | 40.15 | 73.08 | 83.88 | 96.27 | 72.10 | 88.27 | 95.93 | 72.59 |
| | Re-ranking (Liu et al.) | 50.55 | 81.75 | 89.78 | 97.18 | 80.04 | 91.90 | 96.58 | 80.90 |
| | CaLa (Jiang et al., 2024) | 49.11 | 81.21 | 89.59 | 98.00 | 76.27 | 91.04 | 96.46 | 78.74 |
| | CCIN (Tian et al., 2025) | 53.41 | 84.05 | **91.17** | 98.00 | - | - | - | - |
| | QuRe (Kwak et al., 2025) | 52.22 | 82.53 | 90.31 | 98.17 | 78.51 | 91.28 | 96.48 | 80.52 |
| | SPRC† (Xu et al., 2024) | 52.05 | 82.22 | 89.98 | 97.88 | 80.31 | 91.88 | 96.58 | 81.27 |
| | **+ DAIG (Ours)** | **53.88** | **84.10** | 90.60 | **98.41** | **80.70** | **92.48** | **96.94** | **82.40** |

Table 3: **Results on different datasets.** Experiments are based on SPRC, where *DAIG-32-shot* is our 32-shot first stage results (only DRSP). † indicates that the result is from the original paper.

| Dataset | Size | CIRR Avg. | FashionIQ Avg@10 |
|---|---|---|---|
| ST18M | 18M | 62.47 | 30.97 |
| LaSCo | 389k | 68.29 | 30.81 |
| WebVid-CoVR | 1.6M | 67.81 | 34.37 |
| CIRHS† | 534k | 69.14 | 37.44 |
| **DAIG-32-shot (Ours)** | **20k** | **71.68** | **44.74** |

Table 4: **Comparison under few-shot CIR.**

| Rate | Method | CIRR Recall@5 | FashionIQ Avg@10 |
|---|---|---|---|
| 8-shot | PromptCLIP | 41.0 | 19.9 |
| | SPRC | 55.2 | 25.7 |
| | +PTG | 65.5 | 30.6 |
| | **+DAIG (Ours)** | **69.5** | **43.7** |
| 16-shot | PromptCLIP | 40.8 | 21.2 |
| | SPRC | 61.3 | 26.3 |
| | +PTG | 67.5 | 31.3 |
| | **+DAIG (Ours)** | **70.1** | **44.5** |

## 4.3 ABLATION STUDY

**Comparison with other training datasets.** Since the key of our approach is the plug-and-play domain-adaptive in-context generation paradigm, we conduct ablation study to validate the superiority of our synthetic triplets. As illustrated in Table 3, SPRC trained with our synthetic dataset consistently outperforms those trained with other open-source datasets, including ST18M (Gu et al., 2024a), LaSCo (Levy et al., 2024), WebVid-CoVR (Ventura et al., 2024), and CIRHS (Li et al., 2025) constructed via the zero-shot in-context generation. This indicates that, without target domain knowledge, triplets constructed either from real image-text datasets or via zero-shot in-context generation are biased and suboptimal. However, with the support of few-shot target domain data, our DAIG significantly improves the quality of the constructed triplets, achieving better performance with a size of only 20k, particularly surpassing the zero-shot in-context generated triplet dataset, namely CIRHS, which includes 534k synthetic triplets.

**Effectiveness of different components.** As reported in Table 5, for the DRSP stage, the vanilla SPRC trained on Zero-Shot In-context Generated (ZSIG) triplets lags our DAIG equipped with a standard LoRA (Hu et al., 2021) on both the CIRR test set and the FashionIQ validation set, demonstrating that high-quality domain-adapted triplets significantly benefit CIR model performance. Building on this, our proposed CIR-LoRA yields a further improvement of **1.66%** and

Table 5: **Ablation Study.** The best results are marked in bold. FRA is evaluated on 1% of the human-annotated data.

| Stage | Method | CIRR Recall@K | | | | FashionIQ | |
|---|---|---|---|---|---|---|---|
| | | K=1 | K=5 | K=10 | K=50 | Avg@10 | Avg@50 |
| DRSP | ZSIG | 36.07 | 66.10 | 77.23 | 92.87 | 39.66 | 61.09 |
| | DAIG w/ LoRA | 39.33 | 70.19 | 80.29 | 94.34 | 41.62 | 63.59 |
| | DAIG w/ CIR-LoRA | 40.99 | 71.01 | 81.42 | 94.75 | 43.89 | 65.29 |
| | **DAIG w/ CIR-LoRA+perturb** | **42.05** | **72.02** | **82.00** | **95.25** | **45.02** | **66.38** |
| FRA | w/o Angular Margin $\varphi$ | 43.98 | 74.65 | 84.46 | 95.90 | 44.98 | 66.87 |
| | **w/ Angular Margin $\varphi$** | **44.55** | **75.52** | **84.87** | **96.22** | **45.51** | **67.30** |

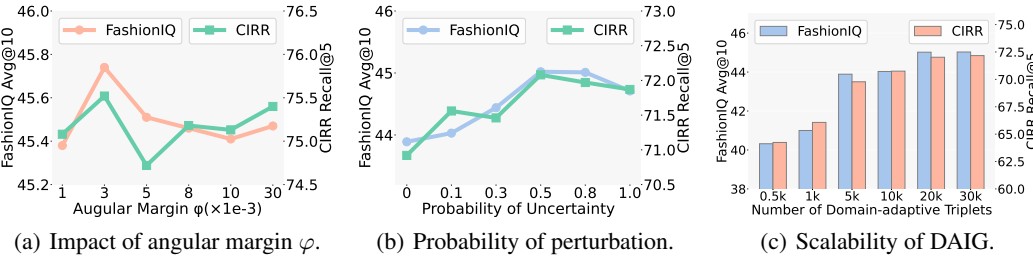

(a) Impact of angular margin $\varphi$.   (b) Probability of perturbation.   (c) Scalability of DAIG.

Figure 3: **Hyperparameter analysis.** Experiments are built upon SPRC+DAIG. (b) and (c) show the results of the first stage (only DRSP), while (a) includes FRA trained with 1% of real-world data.

**2.27%**, highlighting the effectiveness of the tailored design for CIR. Moreover, applying perturbation ($p = 0.5$) leads to an additional gain of **1.06%** and **1.13%**, confirming its role in enhancing the model's robustness in the visual branch. For the FRA stage, we show that the angular margin $\varphi$ plays a critical role in capturing fine-grained information, outperforming no-margin counterpart by **0.87%** on the CIRR Recall@5, and **0.53%** on the FashionIQ average Recall@10.

**Hyper-parameter Sensitivity.** Our method incorporates several key hyper-parameters, including the perturbation probability $p$ in the DRSP stage, the angular margin $\varphi$ in the FRA stage, and the number of synthesized domain-adaptive triplets. We conduct ablation studies to examine the sensitivity of these hyper-parameters. As shown in Figure 3(a), the optimal angular margin $\varphi$ is set to $3 \times 10^{-3}$, while the sensitivity of $\varphi$ is relatively low that our method consistently outperforms existing approaches across different values. For the probability of perturbation $p$ in Figure 3(b), the best performance is observed at $p = 0.5$, which yields significant improvement over the no-perturbation setting $p = 0$, demonstrating the effectiveness of our design. Finally, regarding the scalability of DAIG in Figure 3(c), performance largely improves when increasing the number to 5k, and achieves better performance at 20k, and further scaling to 30k offers only marginal gains.

**Computational Overhead.** Although our method leverages the T2I generation, the computational and time costs are manageable. DAIG fine-tunes Flux on a single H800 GPU in a few hours, and generating a triplet takes just around 3 seconds on average with a resolution of 224×448.

## 5 CONCLUSION AND DISCUSSION

In this work, we propose Domain-Adaptive In-context Generation (DAIG), which injects domain and CIR task priors into a T2I model using few-shot human-annotated triplets, and transforms LLM-generated textual triplets into domain-adaptive CIR triplets with a layout template. To leverage the synthetic and few-shot real-world triplets, we present a two-stage framework that can be seamlessly integrated with any supervised CIR method, leading to significant performance gains in both few-shot and fully supervised settings. Future research directions include multi-resolution reference and target image generation, testing other T2I models, *e.g.*, SD3.5 (Esser et al., 2024) and PixArt (Chen et al., 2023), and exploring step distillation (Yin et al., 2024) for faster DAIG processing. Our method holds great potential, and we believe it will find broader applications in the future.

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

## A  USAGE OF LARGE LANGUAGE MODELS

In our work, LLMs are employed in two primary aspects. First, during the construction of domain-adaptive triplets, we utilize an LLM, specifically Qwen2.5-72B-Instruct (Yang et al., 2025),to generate textual triplets; the corresponding details are provided in Sections 3.2 and B.2. Second, we employ an LLM, namely GPT-5 (Achiam et al., 2024), for translation and refinement of our manuscript. The prompt used is presented as follows,

> ***Polishing Prompt***: Assume you are a senior expert in translation and academic English writing. Please help me translate several paragraphs into English, ensuring that the language meets the standards of professional academic journals. The requirements for the translation are as follows: (1) Ensure that the translated English text is consistent with the meaning of the original, without altering the intended message. (2) Provide precise definitions, ensuring that terminology and technical terms are used accurately, especially for domain-specific concepts. (3) Maintain logical accuracy and clarity of reasoning in the language. (4) Use concise and clear expressions, avoiding vague or unnecessary words or sentences. (5) Pay close attention to grammatical accuracy, ensuring that sentence structures are correct.

## B  DATASET DETAILS

### B.1  BENCHMARKS FOR COMPOSED IMAGE RETRIEVAL

FashionIQ (Wu et al., 2021) is established to advance research on conversational interfaces for online fashion shopping, aiming to move beyond traditional keyword-based retrieval systems that often

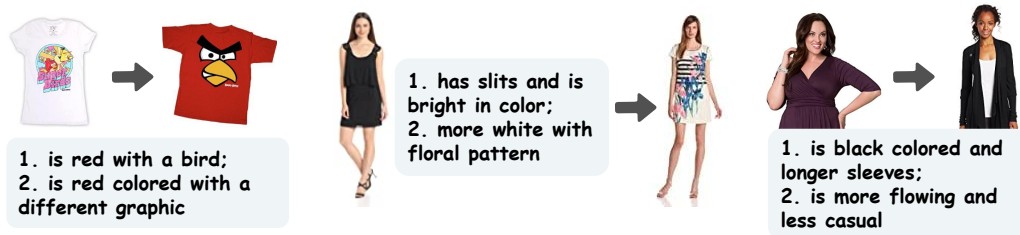

(a) Fashion IQ exampels. Left: Shirt; Middle: Dress; Right: Toptee.

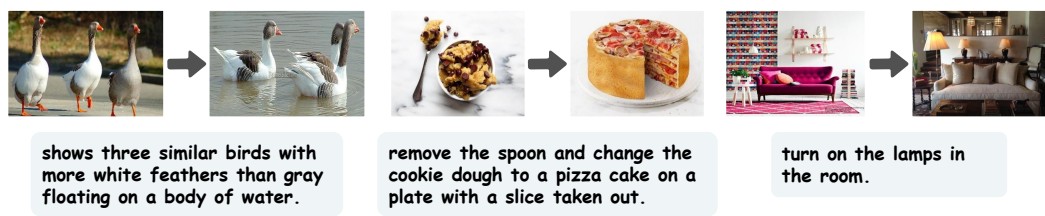

(b) CIRR exampels, covering a wide range of real-life scenarios.

Figure 4: **Examples of CIR benchmarks.** In all instances, the left side represents the reference image, while the right side is the target image obtained through the given relative caption.

fail to capture user intent. The dataset focuses on interactive fashion image retrieval, a canonical form of the CIR task. It consists of 30,134 triplets derived from 77,684 fashion images, organized into three categories: Dress, Shirt, and Toptee. Each triplet contains a reference image, a relative caption describing the intended modification, and a target image. In addition to the triplet structure, the dataset also provides product descriptions and attribute-level annotations, which support more fine-grained evaluation. The official split follows a 6:2:2 ratio for training, validation, and testing. Representative examples are visualized in Figure 4(a).

CIRR (Liu et al., 2021) is proposed to broaden CIR research to open-domain retrieval scenarios, addressing the limitation that most existing benchmarks—such as FashionIQ—are confined to a single domain. CIRR is constructed by first sampling a large set of visually similar natural images from NLVR (Suhr et al., 2019), using ResNet-152 (He et al., 2016) pre-trained on ImageNet (Russakovsky et al., 2015) as the similarity backbone. Pairs of highly similar images are then manually annotated with relative captions to form triplets. In total, CIRR comprises 36,554 annotated triplets, which are randomly divided into training, validation, and testing splits with an 8:1:1 ratio. Evaluation is conducted through a remote server submission system, ensuring fair benchmarking across methods. Despite its strengths, CIRR is not without challenges: some relative captions contain vague or redundant descriptions, and the dataset includes a considerable number of false negatives (FNs), which may adversely affect evaluation accuracy. Visual examples are provided in Figure 4(b).

### B.2 MORE DETAILS ON DOMAIN-ADAPTIVE IN-CONTEXT GENERATION

Here we provide the instruction template involved in DAIG, along with the lists of object and edit, which can be generated using GPT-5 (Achiam et al., 2024), as shown below. More samples generated by DAIG in the fashion domain and real-life scenarios are shown in Figure 6, with the word cloud visualization of their relative captions displayed in Figure 5.

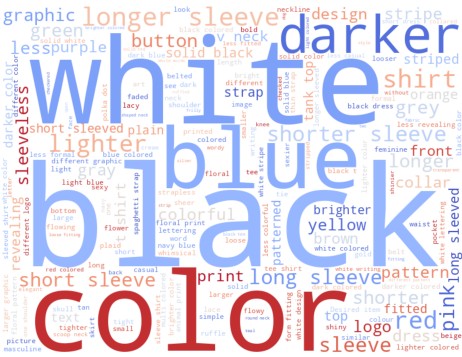
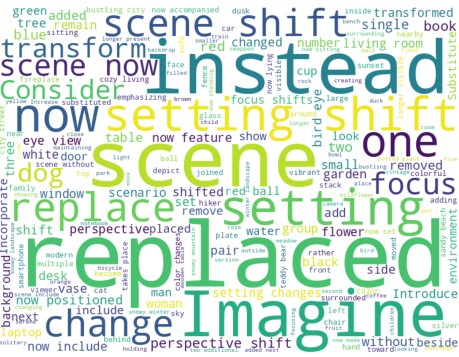

(a) Word cloud visualization of DAIG in the fashion domain.

(b) Word cloud visualization of DAIG under real-life scenarios.

Figure 5: **The relative captions word cloud visualization of triplets generated by DAIG**

***Object List***: T-shirt, acorn, adult, airplane, alarm clock, ambulance, ankle boot, baboon, backpack, bakery, balloon, banana, baseball bat, basketball, bathtub, beach, beach ball, beagle dog, beaker, bear, bed, belt, beret, bernese mountain dog, bicycle, bikini, bird, blouse, boat, book, bookcase, bookshelf, bookshop, boot, border collie dog, border terrier dog, bowl, boy, bridge, broccoli, building, bus, cake, calculator, calendar, camera, canoe, cap, car, cat, chair, cheetah, chess set, chicken, chimpanzee, clock, cloud, clownfish, coat, cocker spaniel, collie dog, computer, computer mouse, convertible, cookie, corgi, cow, crowd, cup, curtain, deer, dingo, dining table, doberman dog, dog, dogsled, doll, dolphin, door, dress, drum, duck, dumbbell, elderly person, elephant, envelope, eraser, family, fence, fire truck, fish, flashlight, flower, fork, fox, french bulldog, frog, garden, german shepherd dog, giraffa, girl, glasses, globe, glove, golden retriever, golf ball, goose, gorilla, grass, group of people, guinea pig, guitar, hamster, handbag, hat, headphones, high-heeled shoe, horse, horse cart, hospital, husky, hyena, jacket, jeans, key, keyboard, keypad, kite, knife, labrador retriever, lamp, laptop, printer, lemon, library, lion, lipstick, llama, lock, lotion, mailbox, malamute dog, man, map, marmot, medal, microphone, microwave oven, mobile phone, monastery, money, monitor, monkey, moon, motorcycle, mountain, mouse, museum, notebook, orange, oven, padlock, pajama, pandas, paper towel, park, pelican, pen, pencil, pencil box, penguin, person running, person sitting, person walking, piano, pillow, pizza, plate, police car, poodle dog, potato, pug, puzzle, rabbit, red wine, refrigerator, remote control, restaurant, river, road, rock, rucksack, rug, running shoe, samoyed, sandals, sax, scarf, school, school bus, scissors, sea anemone, sea lion, shark, shawl, sheep, ship, shorts, a sink, skirt, sky, sliding door, slipper, smartphone, sneaker, soccer ball, sock, sofa, spoon, squirrel, stadium, stamp, stingray, subway train, suit, sun, sunglasses, supermarket, sweater, swimming pool, syringe, table, taxi, teddy bear, teenager, telephone, television, tennis racket, tent, theater, tie, tiger, timber wolf, toilet, touchpad, toy, trafic light, train, tree, trifle, tripod, trophy, truck, turtle, umbrella, underwear, vase, vending machine, vizsla dog, vulture, waistband, wallet, warthog, washbasin, watch, whale, window, wolf, woman, yurt, zebra, zoo

***Edit List***: quantity decrease, domain conversion, attribute modification, object composition, orientation change, object addition, object removal, object substitution, quantity change, spatial relation, scenario change, viewpoint shift

***Instruction Template***: You are a visual description expert. You are given a triplet as example: {examples}. Your goal is to carefully analyze the style and structure of this example and generate a new triplet centered around the object category {object}. Follow the three steps below to ensure coherence and consistency. 1) Generate the reference caption: Think carefully about realistic and detailed scenarios involving object. Construct a natural and context-rich sentence that thoroughly describes a scene or moment featuring this object. 2) Generate the edited instruction: Based on the reference caption, imagine a meaningful transformation that focuses on the semantic category {edit}. Write the edited instruction that describes this change clearly. Avoid starting with direct verbs like "add", "change", or "remove". Use fluent, natural sentence structures with more stylistic variety. 3) Generate the target caption: Incorporate the instruction into the original reference caption, generating a informative target caption. Ensure the target caption reflects the transformation clearly while preserving unchanged elements. Note that your output must include only the following content (use double quotes): {"reference caption": ..., "edited instruction": ..., "target caption": ...}. Please only provide the dictionary output, no explanations, no surrounding text.

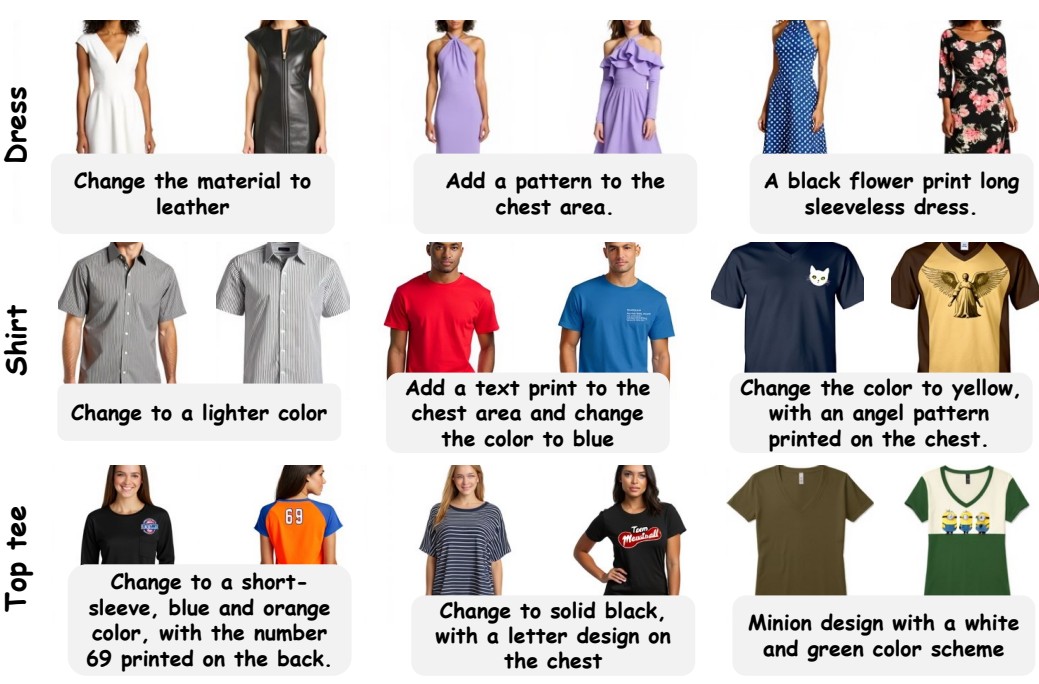

(a) Triplets generated by training DAIG on FashionIQ.

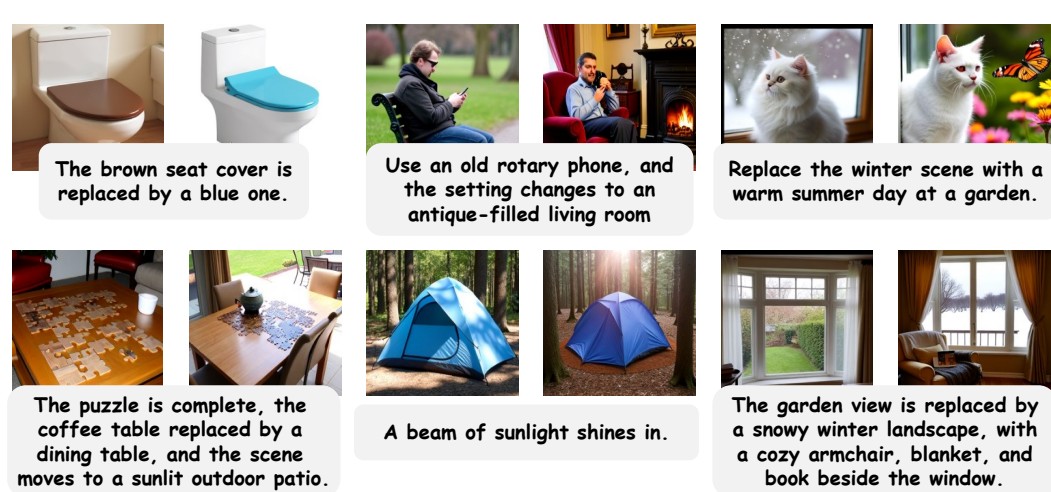

(b) Triplets generated by training DAIG on CIRR in real-life scenarios.

Figure 6: **More triplet examples generated by on DAIG in the fashion domain and real-life scenarios.**