# OpenReview forum: "Domain-adaptive In-context Generation Benefits Composed Image Retrieval"
_ICLR.cc/2026/Conference — ICLR 2026 Conference Withdrawn Submission_

### Official Review · Reviewer_4NAj · 2025-10-21

**Soundness:** 2
**Presentation:** 3
**Contribution:** 2
**Rating:** 2
**Confidence:** 5

**Summary:**

This paper introduces a novel framework for Composed Image Retrieval (CIR) in low-data regimes. The core contribution is Domain-Adaptive In-context Generation (DAIG), a method that fine-tunes a pretrained Text-to-Image (T2I) model on a small set of examples and generates a large-scale, high-quality synthetic dataset of CIR triplets that are well-aligned with the target domain. The authors propose a two-stage training pipeline: 1) Distributionally Robust Synthetic Pretraining (DRSP), which pretrains a CIR model on the synthetic data with feature perturbations for better generalization, and 2) Fine-grained Real-world Adaptation (FRA), which fine-tunes the model on the few available real triplets using an angular margin loss. Experiments show the effectiveness of the proposed method.

**Strengths:**

1. The paper is organized well and is written with clarity, making the complex technical details easy to follow. The figures are effective and easy to understand.

2. Experiments on two widely-used benchmarks verify the proposed method to some extent.

3. This paper proposed a "plug-and-play" method, which allows it to boost the performance of various existing CIR models, making it a valuable contribution to the field.

**Weaknesses:**

1. High System Complexity and Computational Cost: The proposed pipeline is a multi-stage process involving several large-scale models: a VLM for captioning, an LLM for generating textual triplets, and a T2I model for image generation. This is followed by a two-stage training process for the final CIR model. This complexity may present a significant barrier compared to zero-shot methods.

2. More explanation and experiments should be given to elaborate the rationale of the method. This paper does not sufficiently explain how FRA directly addresses or facilitates fine-grained learning, which suggests an ability to interpret subtle semantic details. The paper asserts this outcome but provides no qualitative or quantitative analysis to demonstrate that the model is indeed learning these nuanced distinctions as a result of the angular margin, rather than just learning a more compact embedding space in general.

**Questions:**

1.	Generalization across different T2I Models: The experiments rely on the Flux.1-dev model. Is the approach's effectiveness tightly coupled to the capabilities of the latest-generation T2I models, or do authors believe the principle of domain adaptation would yield significant benefits even with weaker generators?

2.	Computational Overhead. The DAIG pipeline involves multiple complex stages. Could the authors provide a more detailed breakdown of the computational cost for the data generation process? For example, what is the total time and GPU resources needed for fine-tuning the T2I model, generating textual triplets with the LLM, and synthesizing the final triplets? Furthermore, what is the additional training overhead when applying the two-stage framework to an existing CIR model?

3.	Sensitivity to Few-Shot Data: How were the 32 samples for T2I model fine-tuning selected? Have the authors analyzed the performance variance when using different, randomly selected sets of 32 samples? How robust is the generator's adaptation to the choice of these few examples?

---

### Official Review · Reviewer_48BK · 2025-10-30

**Soundness:** 2
**Presentation:** 2
**Contribution:** 2
**Rating:** 4
**Confidence:** 4

**Summary:**

The paper introduces a method for Composed Image Retrieval (CIR)—retrieving a target image given a reference image and a text query. It proposes Domain-Adaptive In-Context Generation (DAIG), which adapts a pretrained text-to-image (T2I) model via Mixture-of-Experts (MoE) LoRA using small (ex. 32) samples to synthesize CIR training triplets (query image, query text, target image). After generating this synthetic data using the data generation pipeline, the approach pretrains a CIR model using the synthetic data with visual feature perturbations, then, fine-tunes it on manually annotated triplets using an angular-margin loss on matching pairs. Given the same samples (even with full supervision setting) for synthetic generation model training, the proposed CIR model's retrieval performance is better than prior works.

**Strengths:**

Under comparable settings, the method achieves better retrieval performance than prior CIR approaches. The gains are largely attributable to (i) domain adaptation of the image-generation pipeline via MoE-LoRA, (ii) perturbation in pretrain step and (iii) the use of an angular-margin loss in the final fine-tuning stage.

**Weaknesses:**

- Limited novelty: the approach largely heuristically combines existing methods
    - The core idea of Domain-Adaptive In-Context Generation (DAIG) appears to follow this paper: https://arxiv.org/pdf/2504.20690
    - CIR_LoRA (MoE LoRA) is similar to this work: https://arxiv.org/pdf/2403.11549
    - The CIR training pipeline uses a standard two-stage setup: pretraining on synthetic data, then fine-tuning on real data. Perturbation is standard way when pre-training with synthetic data and angular-margin loss is similar to https://arxiv.org/abs/1708.01682

-  Evaluation against CompoDiff, VISTA, and CoAlign is needed, as these methods follow the same paradigm of leveraging T2I models for triplet construction.

- The paper is hard to follow; methodological details are not clearly explained in details.

**Questions:**

- Is 32 instances enough for updating MoE LoRA on image generation model? What is the number of parameters to train in the model including routing MLP? How was the weight in MoE LoRA initialized?

---

### Official Review · Reviewer_9LgD · 2025-11-01

**Soundness:** 2
**Presentation:** 2
**Contribution:** 3
**Rating:** 4
**Confidence:** 3

**Summary:**

The paper identifies the problem of domain gaps that existing composed image retrieval (CIR) methods do not focus on. To tackle this, the authors propose Domain-Adaptive In-Context Generation (DAIG), which fine-tunes text-to-image (T2I) models using only a few-shot set of triplets to achieve domain adaptation. Building on this, the paper introduces a two-stage CIR training pipeline (Distributionally Robust Synthetic Pretraining and Fine-Grained Real-World Adaptation) to further enhance generalization. Experimental results demonstrate substantial improvements under few-shot learning settings, and concrete analyses, including hyperparameter ablations, validate the effectiveness of the method.

**Strengths:**

1. The paper is well-written and clearly structured, making it easy to follow.
2. The proposed method shows strong effectiveness in the few-shot CIR setting.
3. The Domain-Adaptive In-Context Generation (DAIG) framework and the two-stage adaptation process (Distributionally Robust Synthetic Pretraining followed by Fine-Grained Real-World Adaptation) are well-motivated and conceptually sound.
4. Extensive experiments across diverse settings further validate the effectiveness of the proposed approach.

**Weaknesses:**

1. The main concern lies in the problem setting. As far as I know, I believe the few-shot CIR setting is a non-representative setup yet.
While few-shot CIR is a potentially meaningful direction, I am not fully convinced that the current benchmark and experimental setup are valid representations of real-world few-shot scenarios. In practical use cases, domain gaps in images tend to be much larger (for instance, transferring from natural to medical or industrial images), whereas the benchmarks used here (CIRR, CIRCO, and FashionIQ) both largely consist of natural or fashion-related images with relatively small domain differences.

Moreover, regarding the use of the T2I model, it seems that the generative model is already well-trained on these domains. Thus, describing the method as “reducing the domain gap” may be somewhat misleading; it might be more accurate to frame it as fitting to a particular dataset.

In this line, the motivation for generating a small subset of synthetic triplets specifically for the few-shot CIR setting feels somewhat unclear. I would appreciate further clarification on the concrete motivation for this choice and how it meaningfully differs from existing data augmentation or domain-adaptive generation methods. It would also be valuable to hear how other reviewers interpret this aspect.


2. While the ablation on DRSP is relatively sufficient, the overall analyses on the two-stage pipeline (DRSP and FRA) are limited. More detailed comparisons with naïve baselines are necessary. For example, DRSP should be compared with a baseline that does not apply visual perturbation to isolate its true contribution.
Additionally, the angular margin introduced in FRA is not well-motivated in the paper. It would be helpful to clarify the intuition behind this design choice and whether alternative formulations were considered or tested. Such comparisons would make the contribution of FRA more convincing.

Now, I'm leaning toward more weak rejection, but I want to hear more about the other reviewer's thinking and author's responses.

**Questions:**

Wrote above

---

### Official Review · Reviewer_r8tK · 2025-11-04

**Soundness:** 3
**Presentation:** 3
**Contribution:** 2
**Rating:** 4
**Confidence:** 4

**Summary:**

This paper proposes the domain-adaptive in-context generation (DAIG) method for the task of composed image retrieval, which adapts the in-context capability of pretrained Text-to-Image models to the target domain and the CIR task with few-shot samples, transforming the LLM-generated textual triplets into unbiased CIR triplets as additional training data.

**Strengths:**

1. The paper is well written, and the methodology is clearly explained.
2. Extensive ablations results demonstrate the effectiveness of the proposed method.

**Weaknesses:**

1. The motivation is not clear. The paper targets at zero-shot methods' intractable domain gap without manually annotated data, while the proposed method still utilizes labeled data. The targeted problem and the proposed method appear to be inconsistent.
2. The proposed architecture is complex, and no efficiency evaluation results are provided. Is the proposed method practival in real-world applications?

**Questions:**

1. More clarification on motivation concerning labeled / unlabled data should be presented, inproving the consistency between the motivation and the proposed method.
2.  Explicit comparison results of computational overhead should be presented in tables, instead of simple descriptions within a few lines.
3. Efficiency evaluation results including inference time should be provided.

---

### Note · Authors · 2025-11-14

I have read and agree with the venue's withdrawal policy on behalf of myself and my co-authors.